# Friendly Noise against Adversarial Noise: A Powerful Defense against Data Poisoning Attacks

**Tian Yu Liu**
Department of Computer Science
University of California, Los Angeles
tianyu@cs.ucla.edu

**Yu Yang**
Department of Computer Science
University of California, Los Angeles
yuyang@cs.ucla.edu

**Baharan Mirzasoleiman**
Department of Computer Science
University of California, Los Angeles
baharan@cs.ucla.edu

## Abstract

A powerful category of (invisible) data poisoning attacks modify a subset of training examples by small adversarial perturbations to change the prediction of certain test-time data. Existing defense mechanisms are not desirable to deploy in practice, as they often either drastically harm the generalization performance, or are attack-specific, and prohibitively slow to apply. Here, we propose a simple but highly effective approach that unlike existing methods breaks various types of invisible poisoning attacks with the slightest drop in the generalization performance. We make the key observation that attacks introduce local sharp regions of high training loss, which when minimized, results in learning the adversarial perturbations and makes the attack successful. To break poisoning attacks, our key idea is to alleviate the sharp loss regions introduced by poisons. To do so, our approach comprises two components: an optimized friendly noise that is generated to maximally perturb examples without degrading the performance, and a randomly varying noise component. The combination of both components builds a very light-weight but extremely effective defense against the most powerful triggerless targeted and hidden-trigger backdoor poisoning attacks, including Gradient Matching, Bulls-eye Polytope, and Sleeper Agent. We show that our friendly noise is transferable to other architectures, and adaptive attacks cannot break our defense due to its random noise component. [1]

## 1 Introduction

Big datasets empower modern over-parameterized deep learning systems. Such datasets are often scraped from the internet or other public and user-provided sources. An adversary can easily insert a subset of malicious examples into the data collected from public sources to harm the model's behavior at test time. As a result, deep learning systems trained on public data are extremely vulnerable to data poisoning attacks. Such attacks modify a subset of training examples under small (and potentially invisible) adversarial perturbations, with the aim of changing the model's prediction on specific test-time examples. Powerful attacks generate poisons that visually look innocent and are seemingly properly labeled [10, 15, 34]. This makes them hard to detect even by expert observers. Hence, data poisoning attacks are arguably one of the most concerning threats to modern deep learning systems [19].

---

[1]Our code can be found at https://github.com/tianyu139/friendly-noise

36th Conference on Neural Information Processing Systems (NeurIPS 2022).

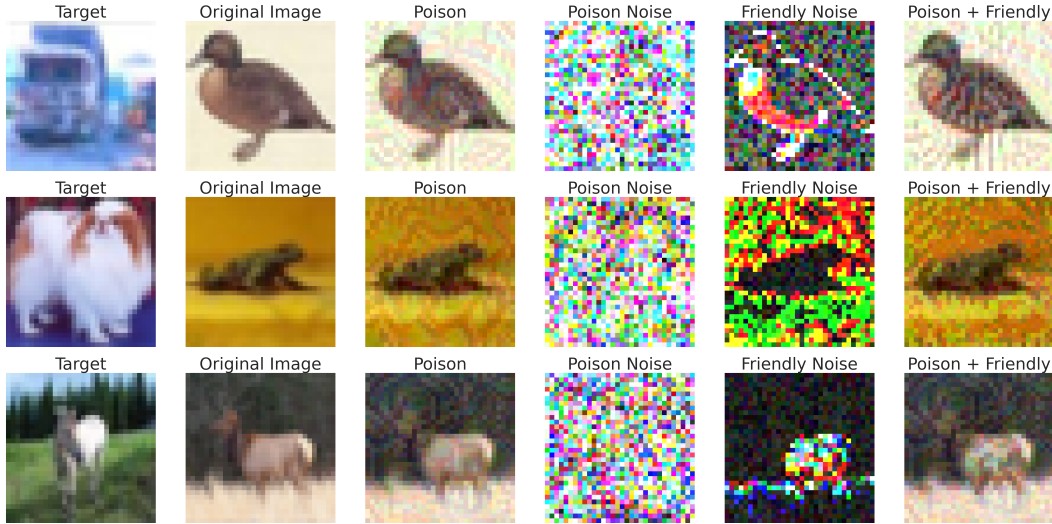

Figure 1: Qualitative Evaluation of Friendly Noise. Our optimized noise adds maximum allowed perturbation to the regions where network robustly learns and leaves other areas untouched (darker regions means less noise).

Various types of poisoning attacks have been proposed to challenge and exploit the vulnerabilities of deep learning systems. Backdoor data poisoning attacks add a fixed but not necessarily visible trigger pattern to a subset of training data as well as the test-time target examples [13, 34, 39]. Triggerless poisoning attacks add small bounded perturbations to a subset of training examples to make them similar to the adversarially labeled test-time target in the feature or gradient space [2, 12, 15, 32, 44]. In both cases, training or fine-tuning the model on the poisoned training data causes the model to misclassify certain target examples at test time.

There have been sustained efforts to design effective defense mechanisms [1, 7, 11, 17, 25, 27, 33, 38, 43]. However, existing methods are highly impractical to be employed in real-world deep learning pipelines. Firstly, the majority of the existing methods are attack specific and cannot protect the system against various types of data poisoning attacks [11, 27, 38]. Secondly, the provided protection is often at the expense of significantly dropping the performance of the machine learning pipeline [1, 7, 25]. Thirdly, existing methods are not effective in protecting the deep learning pipelines against adaptive attacks which can make more powerful poisons with the knowledge of the defense in place [17, 33]. Finally, state-of-the-art defense methods are often so expensive that they can hardly be applied to even medium-sized datasets [11, 27], and are ineffective in presence of larger number of poisons [7, 11, 27].

In this work, we propose a simple and powerful defense, namely Friendly Noise Defense (FRIENDS), against various types of visually imperceptible data poisoning attacks. In particular, we make the following key observation: data poisoning attacks introduce local sharp regions with high training loss by adding adversarial perturbations to a subset of training examples. To effectively break poisoning attacks, our proposed method is composed of two noise components. First, we find the maximum perturbation that can be added to every example without considerably changing the model's output. This fixed accuracy-friendly perturbation is found early in training and is transferable to other architectures. Then, we add a small varying random noise in addition to the friendly perturbation to each example at every training iteration. Effectively, the two components alleviate the local high training loss regions introduced by the poisons, and do not allow the attacks to be successful. Despite being very lightweight, FRIENDS can effectively protect deep learning systems against various types of (invisible) poisoning attacks, with a minimum drop in the generalization performance.

We note that the random noise component of FRIENDS makes it extremely difficult for an adaptive attacker to break our defense. Adaptive attacks can bypass defenses by taking the defense mechanism into account when generating poisons. For FRIENDS, while an attacker may use the knowledge of the optimization procedure to bypass the friendly noise component, they need to take into account a prohibitively large number of random noise combinations when generating attacks. This makes

it extremely difficult for the attacker to ensure the effectiveness of an attack in presence of our FRIENDS defense.

Through extensive experiments, we show that our light-weight method renders state-of-the-art visually imperceptible poisoning attacks, including Gradient Matching [10], Bullseye Polytope [2], Feature Collision [32], and Sleeper Agent [34] ineffective, with only a slight decrease in the performance. We also show that the optimized noise component generated based on a particular architecture can be applied to defend other architectures against data poisoning attacks. Therefore, it is easy to apply FRIENDS to real-world deep learning pipelines with minimal additional costs.

## 2  Related Work

**Targeted Data Poisoning Attacks.** Data poisoning attacks on deep networks have been explored along two directions - triggered and triggerless attacks. Triggered attacks, or backdoor attacks, aim to misclassify samples containing a 'trigger' patch as a pre-determined target class during inference time. In the transfer learning or finetuning setting, earlier works [8, 13, 23] relied on label modifications or unbounded image perturbations. These attacks are, however, easy to detect. Subsequently, [30, 34, 39] introduced clean-label and visually imperceptible backdoor attacks. Recently, [34] proposed the first clean-label hidden backdoor attack that is effective on victim models trained from scratch. Triggerless data poisoning attacks aim to misclassify a given target as a pre-determined adversarial class by adding optimized bounded perturbations to a subset of training examples. Such attacks either optimize for feature matching [2, 32, 44] or gradient matching [10] between poisoned and target images, or use meta-learning to solve the poisoning problem directly via bilevel optimization [15] .

**Defense Strategies.** Existing defenses against data poisoning can be divided into filtering and robust training methods. Filtering methods detect outliers in feature space using thresholding [35] and nearest neighbors [27], or activation space [7], or through decomposition of the feature covariance matrix [38]. These defenses typically assume that only small subsets of the data are poisoned, hence removing such points does not significantly harm generalization. In practice, this assumption may not hold, and such defenses can be easily broken by increasing the number of poisons. Moreover, such methods increase training time by orders of magnitudes, as the filtering step requires training the model with poisons, followed by (usually expensive) filtering, and model retraining [7, 27, 35, 38]. Very recently, [43] proposed an efficient method, to iteratively drop examples with isolated (outlier) gradients. In comparison, our method is faster, easy to apply, and transferable to other architectures. Hence, it is suitable for deployment in real-world deep learning pipelines.

Robust training methods apply randomized smoothing [41], strong data augmentation [4], or model ensembling [21]. Other methods impose constraints on gradient magnitudes and directions [14], detects and removes poisons with gradient ascent [22], or apply adversarial training [11, 25, 37]. Deferentially private (DP) training methods have also been explored to defend against data poisoning [1, 5, 16]. Robust training techniques usually involve a significant trade-off between generalization and poison success rate [1, 14, 22, 25, 37], or are computationally very expensive [11, 25]. Compared to augmentation-based and adversarial training methods, our method is simple, fast, and maintains good generalization performance. Compared to data augmentation, the random noise component of FRIENDS is considerably more effective in smoothing the loss landscape, due to its much larger space of independent pixel-level transformations.

**Random and Adversarial Noise.** It is shown that small perturbations can result in large changes in the output of a deep network [36]. Hence, the application of random and adversarial noise has been studied in various domains. In particular, [28] used Gaussian noise to defend against query-based black box attacks, and [29] showed that additive augmentations of Gaussian or Speckle noise is a simple yet very strong baseline for robustness against image corruptions. The application of optimized noise has been mainly studied in the context of adversarial training. [6] used Generative Adversarial Networks (GANs) to generate adversarial perturbations, and [24] relied on meta-learning to learn a noise generator to defend against adversarial perturbations. Moreover, [26, 42] demonstrated the transferability of adversarial perturbations across architectures and domains. In data poisoning, small random noise generated from a particular distribution has been shown to be ineffective for breaking attacks and harmful to the generalization performance [11]. In contrast, we show that random noise combined with our proposed noise optimization approach, can make a highly effective defense mechanism against data poisoning attacks and achieve a superior generalization performance.

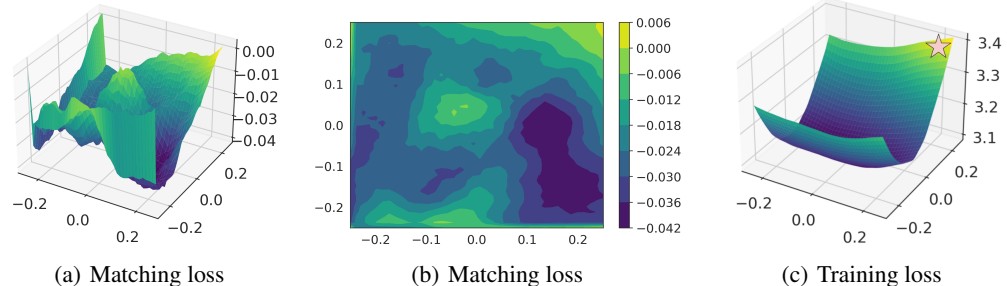

| (a) Matching loss | (b) Matching loss | (c) Training loss |

Figure 2: (a) Matching loss defined as $1-cos\left(\nabla\mathcal{L}(x_t, y_{adv}, \theta), \nabla\mathcal{L}(x_i + \delta, y_i, \theta)\right)$, as we vary the perturbation $\delta_i$ projected along two randomly chosen directions. (b) Contour view of (a). (c) Training loss of a single example $x_i + \delta_i$ as we vary $\delta_i$, projected along two randomly chosen directions. Adding $\delta_i^*$ which minimizes the matching loss to $x_i$ (located at origin), introduces a local region of high training loss (indicated by a star at $x_i + \delta_i^*$).

## 3   FRIENDS: Friendly Noise Defense against Data Poisoning Attacks

Targeted data poisoning attacks modify a fraction of training data points by adding optimized perturbations that are within an $l_\infty$-norm $\xi$-bound. The optimization is done with the objective of changing the prediction of a target example $x_t$ in the test set, to an adversarial label $y_{adv}$. A small perturbation bound $\xi$ ensures that the poisoned examples remain visually similar to the original (base) training data points. Poisons crafted by such attacks look innocent to human observers and are seemingly labeled correctly. Hence, they are called clean-label attacks. Targeted clean-label data poisoning attacks can be formulated as the following bilevel optimization problem:

$$\min_{\delta \in \mathcal{C}} \mathcal{L}(x_t, y_{\text{adv}}, \theta(\delta)) \quad s.t. \quad \theta(\delta) = \arg\min_\theta \sum_{i \in V} \mathcal{L}(x_i + \delta_i, y_i, \theta), \tag{1}$$

where $\mathcal{C} = \{\delta \in \mathbb{R}^{n \times m} : \|\delta\|_\infty \leq \xi, \delta_i = 0 \ \forall i \notin V_p\}$ is the constraint set defining the set of valid poisons, $V$ is the training data, and $V_p$ is the set of poisoned training examples. To address the above optimization problem, powerful poisoning attacks such as Meta Poison (MP) [15], Gradient Matching(GM) [10], Bull-eyes Polytope (BP) [2], and Sleeper Agent [34] craft the poisons to mimic the gradient (equivalently representation in transfer learning) of the adversarially labeled target, i.e.,

$$\nabla\mathcal{L}(x_t, y_{\text{adv}}, \theta) \approx \frac{1}{|V_p|} \sum_{i \in V_p} \nabla\mathcal{L}(x_i + \delta_i, y_i, \theta), \tag{2}$$

Minimizing the training loss on RHS of Eq.(1) also minimizes the adversarial loss on LHS of Eq. (1).

### 3.1   Powerful Poisons Introduce a Local Sharp Region with High Training Loss

Based on Eq. (2), we make the following observation. To substantially change the gradient of a training example $x_i$ to match the adversarial gradient, i.e., $\nabla\mathcal{L}(x_t, y_{\text{adv}}, \theta) \approx \nabla\mathcal{L}(x_i + \delta_i, y_i, \theta)$, under bounded perturbation $\|\delta_i\|_\infty \leq \xi$, the attacker needs to exploit the highly non-convex nature of the loss. That is, the attacker needs to find regions in a ball of radius $\xi$ around example $x_i$, for which $\nabla\mathcal{L}(x_i + \delta_i, y_i, \theta)$ is considerably different than $\mathcal{L}(x_i, y_i, \theta)$. Fig. 2(a), 2(b) illustrate the *matching loss* between the gradient of the adversarial loss and the gradient of a perturbed training example, as we vary the perturbation $\delta_i$ projected along two randomly chosen directions. The matching loss is defined as $1-cos\left(\nabla\mathcal{L}(x_t, y_{adv}, \theta), \nabla\mathcal{L}(x_i + \delta, y_i, \theta)\right)$, where $cos(u, v)$ is the cosine similarity between vectors $u$ and $v$. We see that the adversarial perturbation $\delta_i$ can be effectively optimized to minimize the matching loss in the darker valleys around $x_i$. Such valleys do not exist around all the examples, and hence not every example can be perturbed in a ball of radius $\xi$ to match the adversarial gradient. Indeed, the examples that can be perturbed by $\delta_i$ s.t. $\|\delta_i\| \leq \xi$ to closely match the adversarial gradient are *effective poisons* [43], that make the attack successful. Crucially, each effective poison introduces a local increase to the training loss, as demonstrated by Fig. 2(c). The set of effective poisons together introduce a *local sharp region* with a considerably high *training loss*,

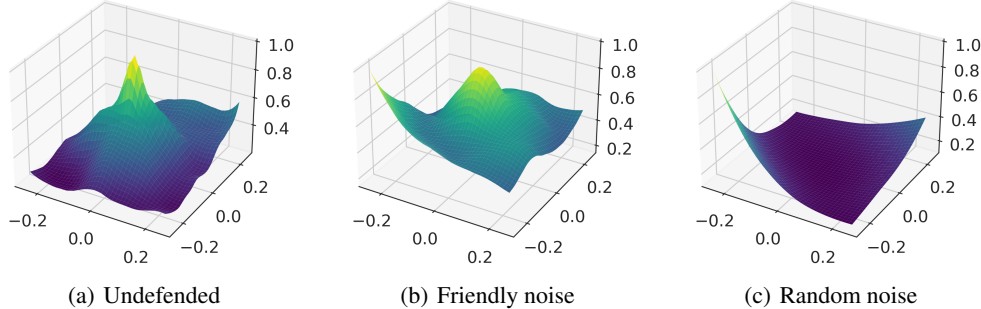

| (a) Undefended | (b) Friendly noise | (c) Random noise |

Figure 3: (a) Loss landscape of a victim model around *all* poisoned examples. Effective poisons introduce a local sharp region with high training loss. (b) Training loss landscape of a victim model defended with friendly noise. (c) Training loss landscape of a victim model defended with random Bernoulli noise. Both components (FRIENDS) smooth the local sharp region introduced by poisons to the training loss (a) and reduce the effectiveness of the attack.

as illustrated by Fig 3(a). Minimizing the training loss on the poisoned data results in learning the adversarial perturbations, and hence a successful attack.

As poisons need to match a particular gradient or representation, they are highly sensitive to small perturbations. In Fig. 5(c) in the Appendix, we show that the model output has indeed a very high standard deviation around the poisoned examples. Therefore, slightly perturbing the poisons considerably changes their gradient and make them ineffective. The main idea behind our friendly noise defense method, FRIENDS, is to maximally perturb the training examples to make the effective poisons ineffective. However, perturbing the training examples should be done in a way that does not harm the generalization performance of the model. To address this, our method is composed of two components: First, we find the maximum perturbation that can be added to every training example without changing its prediction. This alleviates the local sharp region of high training loss introduced by the effective poisons. To further break the attack, we also add a small varying random noise to every example during the training. The random noise generally smooths out the training loss and further alleviates the local sharp regions introduced by effective poisons. Thus, both components together can effectively break the attacks. Below, we discuss each component in more details.

### 3.2  Optimizing the Friendly Noise: Maximally Perturbing Examples without Harm

The first component of our method finds the maximum perturbation that can be added to every example without significantly changing the model's output. To do so, for every example $x_i$ we optimize for the largest noise $\epsilon_i$ within an $l_\infty$ norm $\zeta$-bound that results in similar prediction probabilities, measured by KL-divergence. Formally for each example $x_i$, we find perturbation $\epsilon_i$ as follows[2]:

$$\epsilon_i = \arg\min_{\epsilon:\|\epsilon\|_\infty \leq \zeta} D_{KL}\big(f_\theta(x_i + \epsilon)\|f_\theta(x_i)\big) - \lambda\|\epsilon\|_2. \tag{3}$$

We generate a fixed accuracy-friendly perturbation for every data point by solving problem (3) once, using a few Stochastic Gradient Descent (SGD) steps. There is a trade-off between the time of generating the optimized perturbations and their effectiveness. In particular, the optimized perturbations need to be generated and added to the training examples early in training, before the attack succeeds (note that we aim to prevent the attacks from being successful). At the same time, for the perturbations to be effective, they should be generated after the decision boundary is shaped, so that they can be optimized to minimize the change in the model and its decision boundary. We visualize this trade-off in Appendix B.2. We found that training for as few as 5 epochs before solving the optimization problem (3) yields effective perturbations that reduce the attack success rate without harming the model's performance. The pseudocode can be found in Alg.1.

To better understand the effect of our friendly noise, we illustrate the histogram of the noise added to every pixel in the training data in Appendix B.1, Fig. 4(a). We see that our method mainly

---

[2]We show in Appendix A.1 that using $\|\epsilon\|_1$ or $\|\epsilon\|_\infty$ in Problem (3) yields similar results.

---

**Algorithm 1** Generating Friendly Noise

---

**Require:** Train dataset $X$, Model $f_\theta$, LR $\eta_{opt}$, $\lambda$, small number $T$
    **for** $i \in [T]$ **do**
        $\theta^i = \theta^{i-1} - \eta\nabla_\theta L(\theta^{i-1}, X)$                   $\triangleright$ Train the model for a few epochs
    **end for**
    **for** $x_i \in X$ **do**
        Initialize noise $\epsilon_i^0$ uniformly sampled from $[-\epsilon_{init}, \epsilon_{init}]$
        **for** $t = 1$ to $T$ **do**
            $\epsilon_i^t = \epsilon_i^{t-1} - \eta_{opt}\nabla_\epsilon(D_{KL}\left(f_\theta(x_i + \epsilon_i^{t-1})||f_\theta(x_i)\right) - \lambda\|\epsilon_i^{t-1}\|_2)$
        **end for**
        Store noise $\epsilon_i = \epsilon_i^T$ for example $x_i$
    **end for**

---

---

**Algorithm 2** Training with FRIENDS

---

**Require:** Train dataset $X$, Random Noise Distribution A, Epoch to start defense $def\_epoch$
    Run Algorithm 1 to generate $\{\epsilon_i\}_{i=1}^{|X|}$
    **for** $i = def\_epoch$ to $n\_epochs$ **do**
        **for** $x_i \in X$ **do**
            Sample random noise $\mu_i \sim A$
            Set $\hat{x}_i = x_i + \epsilon_i + \mu_i$ and add to dataset $\hat{X}$
        **end for**
        $\theta^i = \theta^{i-1} - \eta\nabla_\theta L(\theta^{i-1}, \hat{X})$           $\triangleright$ SGD update step with new dataset $\hat{X}$
    **end for**

---

targets certain pixels in every image by adding the maximum amount of perturbation, and leaves the rest of the image untouched. Certain semantic regions have been shown to be much more robust to perturbations [3]. Perturbing the more robust areas does not considerably change the model's behavior and training dynamics. On the other hand, the least robust areas are very sensitive to perturbations, hence small amounts of noise in such areas result in a relatively large change in the model's behavior. We visualize the poisons before and after adding our friendly noise in Fig. 1. We see that our friendly noise successfully targets certain areas in every image that are robustly learned by adding the maximum possible perturbation.

The maximal friendly perturbation added to every example by our method prevents the effective poisons to closely match the target gradient or representation. Effectively, this alleviates the local sharp region of high training loss introduced by the effective poisons, as is illustrated in Fig. 3(b), and considerably reduces the success rate of the attack. At the same time, the important features used for classifications are preserved and hence model predictions remain unchanged. Hence, our method reduces the attack success rate while ensuring only a slight drop in the test accuracy.

Next, we discuss how adding a random variable random noise can further improve the model's robustness against poisoning attacks.

### 3.3 Adding Random Noise: Further Smoothing the Loss

As discussed, our friendly perturbation mainly targets the robust areas of the image and alleviates the local sharp region with high training loss introduced by effective poisons, without considerably changing the training dynamics. However, it does not affect adversarial perturbations that are added to the areas that are not robustly learned. To further improve the model's robustness against poisoning attacks, we add a small variable random noise to all the training examples at every training iteration. Adding the variable random noise generally smooths out the loss landscape. In doing so, it further alleviates the local sharp regions introduced by the attack, and further drops the attack success rate. Note that adding a large random noise can harm the test accuracy by over-smoothing the loss. However, the combination of friendly noise and a smaller random noise can effectively break poisoning attacks without harming the model's performance. Fig. 3(b) and Fig. 3(c) show that the local region of high training loss introduced by effective poisons is alleviated after adding each component of our defense. This clearly demonstrates the effect of our defense and explains its effectiveness. The pseudocode of our defense, FRIENDS, can be found in Alg. 2.

Table 1: Baselines - Against Gradient Matching eps=16, 80 epochs. For trials with all equal outcomes, we report worst-case error estimate $5.59\%$

| Defense | Poison Acc | Test Acc | Time (HH:MM) |
|---|---|---|---|
| AP-0.25 [11] | 15.00% ($\pm 2.85\%$) | 93.27% ($\pm 0.00\%$) | 02:39 |
| AP-0.5 [11] | 10.00% ($\pm 2.01\%$) | 92.83% ($\pm 0.00\%$) | 03:41 |
| **AP-0.75** [11] | **0.00%** ($\pm \mathbf{5.59\%}$) | **91.29%** ($\pm \mathbf{0.00\%}$) | 04:30 |
| DeepKNN [27] | 75.00% ($\pm 4.19\%$) | 93.72% ($\pm 0.24\%$) | 02:55 |
| Adversarial Training [25] | 60.00% ($\pm 5.37\%$) | 92.03% ($\pm 0.31\%$) | 02:37 |
| Activation Clustering [7] | 45.00% ($\pm 5.53\%$) | 87.69% ($\pm 0.50\%$) | 01:01 |
| Diff. Priv. SGD [14] | 5.00% ($\pm 1.06\%$) | 75.70% ($\pm 1.19\%$) | 00:38 |
| **EPIC-0.1** [43] | **0.00%** ($\pm \mathbf{5.59\%}$) | 90.04% ($\pm 0.22\%$) | **00:48** |
| Friendly Noise | 10.00% ($\pm 2.01\%$) | 91.73% ($\pm 0.30\%$) | **00:37** |
| FRIENDS-U | 5.00% ($\pm 1.06\%$) | 91.91% ($\pm 0.28\%$) | **00:37** |
| **FRIENDS-B** | **0.00%** ($\pm \mathbf{5.59\%}$) | **91.52%** ($\pm \mathbf{0.28\%}$) | **00:37** |
| **FRIENDS-G** | **0.00%** ($\pm \mathbf{5.59\%}$) | **91.50%** ($\pm \mathbf{0.25\%}$) | **00:37** |

The small varying random noise can be sampled from various distributions. In particular, we sample from 3 different random noise distributions: (1) Bernoulli: noise is randomly sampled from $\{-\mu, \mu\}$, (2) Uniform: noise is randomly sampled from $[-\mu, \mu]$, and (3) Gaussian: noise is sampled from the normal distribution $\mathcal{N}(0, \mu)$. We note that compared to Uniform noise, Bernoulli and Gaussian noise are in general more effective at reducing the attack success rate, as they make larger perturbations to input features. At the same time, this makes them to suffer a larger drop in the test accuracy. We visualize and compare the Bernoulli, Uniform, and Gaussian noise types in Appendix B.1.

## 3.4 Adaptive attacks

Adaptive attacks can respond to a novel defense algorithm when the attacker is aware of the defense. If the defense algorithm is known to the attacker beforehand, the attacker can generate more powerful poisons by taking into account the specific defense in place. For example, Gradient Matching [10] and Sleeper Agent [34] demonstrated that by including augmented examples as well as original examples during poison generation in Eq. (2), they can obtain robustness against standard data augmentations like crops and flips, when the augmentation technique is preempted by the attacker. For FRIENDS, the prohibitively large search space of random noise permutations and its pixel-wise independence property make it extremely difficult for adaptive attacks to break. That is, while an attacker may use the knowledge of the time and optimization procedure to bypass the friendly perturbation component of FRIENDS, they need to take into account a prohibitively large number of random noise combinations to bypass the random noise component. For example, for a fixed Bernoulli noise, with $p$ pixels there are $2^p$ combinations that an attacker should take into account to ensure the poisons' effectiveness. For real images, this becomes prohibitively expensive. Similarly, for a fixed Gaussian and uniform noise, an infinite number of combinations must be considered during the poison generation to ensure the attack's robustness. Note that our method applies a varying random noise at every iteration, which also needs to be taken into account by the attacker. This makes FRIENDS robust against adaptive attacks.

## 4 Experiments

### 4.1 Implementation details

We evaluate our defense method against both triggerless data poisoning and backdoor attacks under two attack settings - training from scratch and transfer learning. Following the works of [10, 11, 31], we evaluate our method primarily on CIFAR-10, ResNet-18. We also normalize and augment training images with default CIFAR-10 augmentations as used in [11]. For all models trained from scratch, we use a learning rate starting at 0.1 and decaying by a factor of 10 at epochs 30, 50, and 70. For transfer learning, we decay the learning rate at epochs 15, 25, and 35. When applying our method, we clamp the generated friendly perturbations using $\zeta = 16$, and add bounded random noise. For the random noise component, we set $\mu = 16$ in our experiments. We also normalize the image as a

Table 2: Comparisons with Sleeper Agent defenses averaged over 24 datasets (40 epoch setting). ($*$) To accommodate original two-stage learning setup, we ran this for 85 epochs.

| Defense | Poison Acc | Test Acc |
|---|---|---|
| None | 83.48% ($\pm$7.58%) | 91.56% ($\pm$0.19%) |
| Spectral Signatures [38] | 37.17% ($\pm$10.10%) | 89.94% ($\pm$0.19%) |
| Activation Clustering [7] | 15.17% ($\pm$5.38%) | 72.38% ($\pm$0.48%) |
| Diff. Priv. SGD [14] | 13.14% ($\pm$4.49%) | 70.00% ($\pm$0.17%) |
| Strong Augmentation [5] | 69.75% ($\pm$10.77%) | 91.32% ($\pm$0.12%) |
| STRIP [9] | 62.68% ($\pm$4.90%) | 92.23% ($\pm$0.05%) |
| NeuralCleanse [40] | 85.11% ($\pm$5.04%) | 92.26% ($\pm$0.06%) |
| ABL [22] ($*$) | 67.21% ($\pm$10.52%) | 81.33% ($\pm$3.21%) |
| EPIC-0.1 [43] | 23.93% ($\pm$3.48%) | 89.56% ($\pm$0.14%) |
| EPIC-0.2 [43] | **9.10%** ($\pm$**1.57%**) | 86.21% ($\pm$0.14%) |
| **FRIENDS-B** | **21.52%** ($\pm$**8.39%**) | **89.76%** ($\pm$**0.30%**) |
| **FRIENDS-G** | **22.99%** ($\pm$**8.59%**) | **89.87%** ($\pm$**0.31%**) |
| FRIENDS-U | 34.53% ($\pm$9.71%) | 90.36% ($\pm$0.38%) |

Table 3: Against different data poisoning attacks. Here, we use FRIENDS-B as the defense method. Note: Baseline metapoison is ran without default augmentations, following settings used in [11, 15]

| Attack | Scenario | Undefended | | Defended | |
|---|---|---|---|---|---|
| | | Posion Acc | Test Acc | Poison Acc | Test Acc |
| Gradient Matching ($\xi = 8$) | From-scratch | 50.00% | 93.55% | 0.00% | 91.55% |
| Gradient Matching ($\xi = 16$) | From-scratch | 75.00% | 93.50% | 0.00% | 91.52% |
| Metapoison ($\xi = 8$) | From-scratch | 45.00% | 87.61% | 20.00% | 90.82% |
| Bullseye Polytope | Transfer | 100.00% | 92.13% | 35.00% | 79.35% |
| Poison Frogs | Transfer | 100.00% | 92.12% | 30.00% | 79.07% |
| Sleeper Agent | From-scratch | 91.72% | 93.36% | 31.20% | 91.31% |

pre-processing step. We optimize friendly perturbations using SGD with momentum 0.9 and Nesterov acceleration, perform a hyperparameter search along LR= $\{10, 20, 50, 100\}$ and $\lambda = \{1, 10\}$, and optimize each batch of 128 samples for 20 epochs. Following previous works, we report poison success rate (or poison accuracy) as the percentage of datasets poisoned at the end of training. We run all experiments and timings on an NVIDIA A40 GPU.

## 4.2 From-Scratch Setting

First, we evaluate our method on poisoning attacks targeted toward victim models trained from scratch. Such an attack assumes a gray-box scenario, where attackers have knowledge of the victim architecture, but have no knowledge of the specific initialization of the victim's model. Similar to the settings used in [31], which proposes a standardized benchmark for backdoor and data poisoning attacks, benchmark settings, we generate poisoning attacks by selecting $1\%$ of training examples as poisons, which are perturbed within the $l_\infty$ ball of some radius $\xi$. Unless otherwise specified, we set $\xi = 16$. The victim model is initialized with the same architecture targeted by the attack based on a different random seed, and is trained on the poisoned dataset using SGD. When applying FRIENDS, we set $def\_epoch = 5$, and train only with random noise for the first 5 epochs.

### 4.2.1 Baseline Comparison and Ablation Study

We evaluate our method and baseline defenses against the Witches' Brew, or Gradient Matching, attack [10]. It is the current state-of-the-art among data poisoning attacks when applied to the from-scratch setting, and is adapted to be effective against data augmentation and differential privacy [11]. We follow the settings proposed by [11], under which we generate 20 different attack datasets for ResNet-18 trained on CIFAR10 with a $1\%$ budget bounded by $\xi = 16$, with a slight modification - while [11] uses 40 epochs for training, we use 80 epochs to show that our method easily scales to real-world training pipelines. This is because 40 epochs of training only yields $92.01\%$ test error,

Table 4: Ablation study on random noise components of FRIENDS using Gradient Matching attack ($\xi = 16$). We set $\zeta = 32$ for Friendly Noise, and $\mu = 32$ for Noise Only. For experiments on FRIENDS, we set $\zeta = 16$, $\mu = 16$ to combine each component proportionately.

| No Def. | | Friendly Noise | | Noise Type | Noise Only | | FRIENDS | |
|---|---|---|---|---|---|---|---|---|
| P. Acc. | Test Acc. | P. Acc. | Test Acc. | | P. Acc. | Test Acc. | P. Acc. | Test Acc. |
| 75.00 | 93.50 | 10.00 | 91.73 | Gaussian | **0.00** | 89.46 | **0.00** | **91.50** |
| | | | | Bernoulli | 5.00 | 89.31 | **0.00** | **91.52** |
| | | | | Uniform | **0.00** | 91.61 | 5.00 | **91.91** |

while 80 epochs yield $93.50\%$. In Tab. 1, we show that we outperform state-of-the-art defenses [7, 11, 14, 25, 27, 43]. Notably, we achieve the same $0.00\%$ poison success rate with $91.52\%$ test accuracy, an improvement over of $0.23\%$ over state-of-the-art [11] which yield $91.29\%$ test accuracy at the same poison success rate. Most importantly, FRIENDS completes in 37 mins, 7.3x faster than [11] which completes in 4.5hrs. Compared to the efficient method of [43], namely EPIC-0.1 (with $T = 1, K = 5$), FRIENDS achieves a $1.48\%$ higher accuracy while being 1.2x faster. We also strongly outperform other baseline defense methods simultaneously in all three metrics - poison success rate, test accuracy, and runtime. In Tab. 3, we show that FRIENDS also effectively defends against MetaPoison [15], reducing the poison success rate from $45.00\%$ to $20.00\%$ with an accuracy gain from $87.61\%$ to $90.82\%$ resulted from applying augmentations.

We further show that our approach is effective against backdoor attacks, in particular, against the Sleeper Agent attack [34]. Sleeper Agent is the current state-of-the-art clean-label backdoor attack, and the only such attack shown to be effective in from-scratch settings. Following their evaluation protocol, we generate 24 poisoned datasets with $\xi = 16$, and evaluate our defense by training 24 victim models respectively for 40 epochs and testing the poison success rate on 1000 target backdoor images per dataset. We compare our method against other defenses evaluated by [34] in Tab. 2. Here, FRIENDS successfully defends against [34] by reducing poison accuracy from $83.48\%$ to $21.52\%$ with only a small drop in test accuracy from $91.56\%$ to $89.76\%$. We outperform the next best methods, EPIC-0.1 [43] (with $T = 2, K = 5$) and Spectral Signatures [38], by lowering poison accuracy by $14.18\%$ and $2.41\%$ while maintaining similar test accuracy. EPIC-0.2 [43] (with $T = 2, K = 5$) achieves the lowest poison success rate at $9.10\%$, but drops test accuracy to $86.21\%$, and [9] achieves $92.26\%$ test accuracy but suffers from $62.68\%$ poison accuracy.

We also perform an ablation on each components of FRIENDS in Tab. 4. We show that naively applying Friendly Noise ($\zeta = 32$) yields a high poison success rate of $10\%$. On the other hand, applying random noise ($\mu = 32$) yields low poison success rates but also results in a significant test accuracy tradeoff (e.g. $> 4.0\%$ drop for Gaussian and Bernoulli noise). Here, we show that applying FRIENDS by proportionately combining friendly noise ($\zeta = 16$) with each of the random noise components ($\mu = 16$) maintains high test accuracy (i.e. only $2.0\%$ drop) while keeping poison success rate close to 0.

#### 4.2.2 Defending against Adaptive Attacks

As discussed in Sec. 3.4, we believe that an adaptive attack against our defense is computationally prohibitive. To evaluate our claim, we modified the differentiable data augmentation component of Sleeper Agent attack algorithm to include randomly sampled Bernoulli noise. We then further added the fixed friendly noise generated from selected prior runs to the attacker's augmentation procedure. Our results in Tab. 5 show that despite attacks being adapted to Bernoulli and friendly noise, poison accuracies are all within the standard deviation of one another, and the adaptive attacks cannot succeed.

### 4.3 Transfer learning

Next, we evaluate our method on data poisoning and backdoor attacks designed for the transfer learning scenario [2, 32]. Here, the attacks are crafted based on a pretrained network with the goal of achieving poisoning when transfer learning is performed using the generated poisoned dataset. For the transfer learning scenario used in poisoning benchmarks [11, 31], the linear layer (classifier) of the pretrained model is re-initialized and trained with the poisoned dataset, while other layers

Table 5: Defense against Adaptive Attacks generated on Sleeper Agent averaged over 24 datasets (40 epoch setting). Bernoulli and friendly noise are both generated with $\epsilon = 16$.

| Attack | Adaptation | Undefended | | Defended (FRIENDS-B) | |
|---|---|---|---|---|---|
| | | Poison Acc | Test Acc | Poison Acc | Test Acc |
| Sleeper Agent | None | $83.48 \pm 7.58$ | $91.56 \pm 0.19$ | $21.52 \pm 8.39$ | $89.76 \pm 0.30$ |
| Sleeper Agent | Bernoulli Noise | $80.25 \pm 8.13$ | $91.46 \pm 0.27$ | $31.05 \pm 9.45$ | $89.69 \pm 0.33$ |
| Sleeper Agent | FRIENDS-B | $79.08 \pm 8.30$ | $91.62 \pm 0.38$ | $30.50 \pm 9.40$ | $88.42 \pm 0.39$ |

Table 6: Transferability between different architectures

| Method | Poison Acc | Test Acc |
|---|---|---|
| FRIENDS-B (ResNet18) | 0% | 91.52% |
| FRIENDS-B (AlexNet -> ResNet18) | 0% | 91.27% |
| FRIENDS-B (LeNet -> ResNet18) | 0% | 91.39% |

(feature extractor) remain fixed during the training. Similar to the from-scratch setting, attacks are limited to a budget of 1% and $\xi = 16$. However, we generate FRIENDS at the beginning of training instead of after 5 training epochs, since the feature extractor is already initialized. We note that this is not the true transfer learning setting, since the pretraining and transfer learning datasets are the same. However, as [10, 11] noted, this presents an effective worst-case scenario to evaluate poisoning attacks. We show that in Tab. 3 that even in such cases, we reduce poison success rate from 100% to 35% for the Bullseye Polytope attack [2], and from 100% to 30% for the Poison Frogs attack [32].

### 4.4 Transferability across Architectures

We show that perturbations generated by FRIENDS are transferable across architectures. In Tab. 6, we show using Gradient Matching $\xi = 16$ that FRIENDS optimized using smaller architectures, in particular AlexNet [18] and LeNet [20], can be directly used for larger architectures like ResNet18. This presents a significant advantage in terms of computational costs, since FRIENDS can be generated using smaller, and hence faster, models. Crucially, this makes the generated friendly noise free to be directly applied to (much larger) architectures.

## 5 Conclusion

We proposed a simple and highly effective defense mechanism, FRIENDS, that protects deep learning pipelines against various types of poisoning attacks. Our defense is built on the observation that poisoning attacks introduce local sharp regions with high training loss, by adding adversarial perturbations to a subset of training examples. FRIENDS relies on two components to break the poisons: an accuracy-friendly perturbation that is generated to maximally perturb examples without degrading the performance, and a randomly varying noise component. The first component alleviates the local sharp regions introduced by poisons, and the second component further smooths out the loss landscape. Both components combined together build a very light-weight but highly effective defense against the most powerful triggerless and backdoor poisoning attacks, including Gradient Matching, Bull-eyes Polytope, Poison Frogs, and Sleeper Agent, in transfer learning or training from scratch scenarios. FRIENDS is extremely difficult to break with adaptive attacks and our friendly noise can be transferred to other architecture. This makes it almost free to apply to real-world deep learning pipelines. Our defense is particularly targeted towards clean-label poisoning attacks that are generated under bounded perturbations. Such settings are the most difficult to defend, as generated poisons can easily fool even an expert observer. In contrast, unbounded attacks can be easily detected by manual or automated filtering mechanisms, through a single pass over the dataset.

## 6 Acknowledgements

This research was partially supported by Cisco Systems, the National Science Foundation CAREER Award 2146492, and the UCLA-Amazon Science Hub for Humanity and AI.

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
