# Supplementary Material:
# Friendly Noise against Adversarial Noise: A Powerful Defense against Data Poisoning Attacks

## A    Additional Experiments

### A.1    Ablation on different norms for friendly noise objective

Recall the friendly noise objective is given by the following

$$\epsilon_i = \underset{\epsilon:\|\epsilon\|_\infty \leq \zeta}{\arg\min}\, D_{KL}\big(f_\theta(x_i + \epsilon)\|f_\theta(x_i)\big) - \lambda\|\epsilon\|_2. \tag{1}$$

We note that while we used the $L_2$ norm for encouraging larger values of $\epsilon$, other norms such as $L_\infty$ and $L_1$ can also be used. $L_\infty$ only cares about the largest element in the noise vector, so it adds larger perturbations, while $L_1$ cares about absolute values hence it adds smaller (but more non-zero) perturbations. $L_2$ encourages larger noise elements compared to $L_1$. Tab. 1 shows that different norms result in similar poison and test accuracy.

Table 1: Ablation of different norms for the friendly noise objective on CIFAR-10 - Gradient Matching ($\xi = 16$) Performance is similar across L1/L2 norms. Here we use FRIENDs-B with the same defense settings as Tab. 1.

| Norm | Poison Acc | Test Acc |
|:---:|:---:|:---:|
| $L_2$ | 0.00% | 91.52% |
| $L_1$ | 0.00% | 91.50% |
| $L_\infty$ | 0.00% | 91.37% |

## B    Additional Visualizations

### B.1    Histogram visualizations of friendly noise and its variants

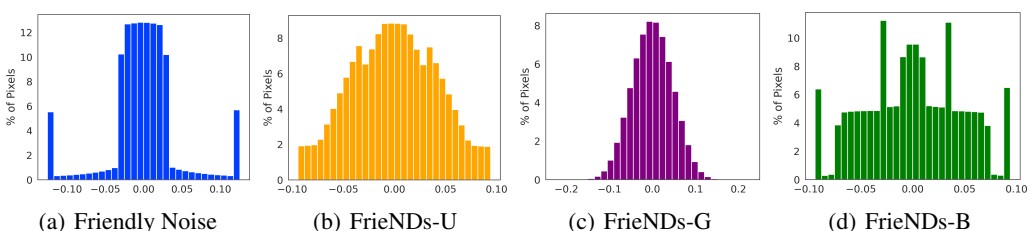

(a) Friendly Noise         (b) FrieNDs-U         (c) FrieNDs-G         (d) FrieNDs-B

Figure 1: Histogram of our method with different types of random noises For (a), we set $\zeta = 32$. For (b)-(d), we set $\zeta = 16, \mu = 16$.

Fig. 1 compares the distribution of random noise sampled from these distributions combined with the optimized perturbation obtained from Eq. (3). We can see that uniform noise perturbs all the pixels similarly, and hence small amount of uniform noise does not harm the model's performance but cannot effectively breaks poisons. Larger uniform noise, however, has a larger effect on the model's performance. Bernoulli and Gaussian noise on the other hand add a larger perturbation to individual pixels. Hence, they are more effective in reducing the attack success rate, but they harm the test accuracy more as they the larger perturbation may be added to the more sensitive areas. Figures 1(b) to 1(d) shows the distribution of random noise combined with our optimized noise added to different pixels. We observe that random noise is added to regions where friendly noise is less dominant, hence resulting in a significant perturbation to an overall greater number of pixels to break poisoning attacks.

## B.2 Trade-off between the time of generating friendly noise and their effectiveness

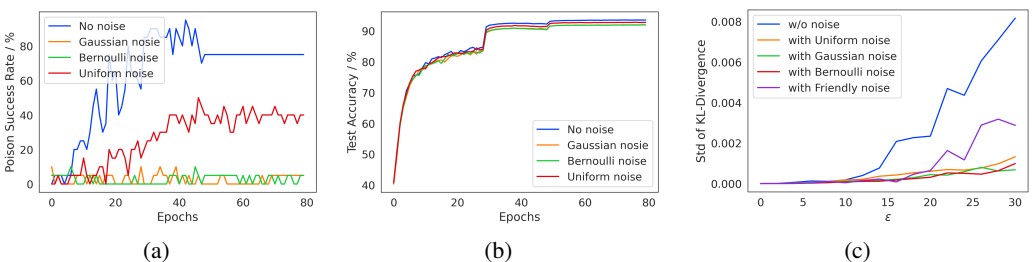

Figure 2: (a) Number of poisoned datasets vs epochs. It takes at least a few epochs for the poisons to have an effect. (b) On the other hand, generalization of the predictions and features learnt increases over time, as measured by error on the test set. (c) Standard deviation of KL-divergence (y-axis) between predictions of training examples before and after adding friendly perturbations in Eq. (3) and random noise sampled from various distributions. Standard deviation is calculated over 10 randomly sampled points in an $\xi$ balls (x-axis) around every training example.