# OpenReview forum: "Friendly Noise against Adversarial Noise: A Powerful Defense against Data Poisoning Attack"
_NeurIPS.cc/2022/Conference — NeurIPS 2022 Accept_

### Official Review · Reviewer_Nmev · 2022-07-09

**Rating:** 5
**Confidence:** 4
**Soundness:** 3 good
**Presentation:** 3 good
**Contribution:** 2 fair

**Summary:**

This paper considers targeted data poisoning attack (in short, we want to classify x as label y_adv, we feed x+\delta with right label y, but the trained network will classify the original x into targeted wrong label).

This paper considers an interesting approach of using optimization to add friendly noise so as to bring the training outside of the sharp loss region. Further the friendly noise is a randomized mechanism, which makes adaptive attacks harder to apply

**Questions:**

Is there actual evidence that adaptive attacks are hard to apply, beyond a heuristic argument of "words" about exhaustive search?

**Limitations:**

No limitation identified

**Strengths And Weaknesses:**

Strengths: The approach is clearly motivated and well executed. The paper is well written and the experimental results seem convincing.

Weakness: The paper did an argument to say that "adaptive attacks is impossible", this may be a little weak -- because one can still think of adaptive attack schemes. For example, how about the adversaries trying to simulate a few such training with different randomness, and try to find poison that breaks all these training? While this seems to be time-consuming, it seems to be a plausible attack scheme.  Note that this may not be the case that the adversaries have to try ALL randomness, since the random Gaussian noise are all centered at mean, one may only need a few samples, and "breaking all of them" will ensure transfer. Nevertheless these are very data dependent, so without experiments or careful thinking it's hard to conclude

---

> ### Author Response · Authors · 2022-08-02
> **Response to Reviewer Nmev**
>
> We thank the reviewer for the valuable feedback and acknowledging our interesting approach to optimize noise to defend against targeted data poisoning attacks.
>
> **Adaptive attacks**: We believe that an adaptive attack against our defense is *extremely difficult* due to the following reasons (we modified the manuscript accordingly to reflect the reviewer’s suggestion). First, note that it is nontrivial, or even almost computationally impossible, for an attacker to be fully adaptive to our friendly noise. Adaptively generated poisons would have to be optimized based on the friendly noise generated. But, the friendly noise generated will in fact be a function of the adaptively generated poisons. In other words, any adaptive poisoned dataset $D_{ap}$ has to be a function of itself, i.e. we require $D_{ap} = g(f(D_{ap}))$ where $f(.)$ gives the friendly noise and $g(.)$ generates the attack. Considering that $f(.)$ consists of many steps of SGD, and $g(.)$ itself is usually equally complex, directly optimizing for such a dataset is computationally prohibitive.
>
> Second, note that the second part of our method, i.e. *random varying noise*, prevents any adaptive attack from being successful even with the knowledge of the fixed noise (as discussed in Sec 3.4). Adaptive attacks incorporate an existing defense in the poison generation process (Eq. 1, 2). However, as our method adds *random varying noise* (that is changing for every example at every iteration) to the examples, it becomes prohibitively expensive for an adaptive attacker to incorporate all different combinations of random noise (in addition to the fixed noise) while crafting poisons to adaptively attack our defense.
>
> We ran new experiments showing that incorporating random noise in addition to the fixed noise (assuming the attacker is aware of it) during poison generation cannot fully break our defense method. We modified the differentiable data augmentation component of Sleeper Agent attack algorithm to include randomly sampled Bernoulli noise of $\epsilon=16$ (the same as what we use in our defense) in row 2. In row 3, we further added the fixed friendly noise generated from selected prior runs to the attacker’s augmentation procedure. Due to time constraints, we only ran 10 datasets for rows 2-3, we will update with the full 24 datasets. We observe that despite attacks being adapted to Bernoulli and friendly noise, poison accuracies are all within standard deviation of one another, and the adaptive attacks cannot succeed. We note that Sleeper Agent attacks (and hence the adaptive attacks in rows 2-3) are by default generated using multiple reinitializations.
>
>
> Table: Sleeper Agent attack (“adaptive”) generated with random Bernoulli noise, with ξ = 16 (averaged over 10 datasets). Even though attack is generated “adaptively” with Bernoulli $\epsilon=16$ noise, it still fails to be robust against Bernoulli noise defense.
> ||Undefended|         |Defended (FRIENDs-B)|     |
> |---|---|---|---|---|
> |Attack|Poison Acc|Test Acc|Poison Acc|Test Acc
> Sleeper Agent|$83.48 \pm 7.58$|$91.56 \pm 0.19$|$21.52 \pm 8.39$|$89.76 \pm 0.30$
> Sleeper Agent + Bernoulli Noise|$81.30 \pm 12.33$|$91.40 \pm 0.21$|$29.57 \pm 14.43$|$89.77 \pm 0.41$
> Sleeper Agent + Friendly and Bernoulli noise|$77.18 \pm 13.27$|$91.71 \pm 0.33$|$31.56 \pm 14.70$|$88.48 \pm 0.41$

---

> ### Author Response · Authors · 2022-08-06
> **Looking forward to the reviewer response**
>
> Since we are approaching the end of the author/reviewer discussion period, we appreciate it if the reviewer can read our clarifications and let us know if there is any remaining point we can clarify further. We believe our new results and clarifications address **all the concerns** and we look forward to any further discussion.

---

### Official Review · Reviewer_tsU4 · 2022-07-09

**Rating:** 6
**Confidence:** 3
**Soundness:** 4 excellent
**Presentation:** 4 excellent
**Contribution:** 3 good

**Summary:**

In this paper, the authors propose a new defense against data-poisoning attacks. The authors train the models on the data perturbed with 2 types of noise – (1) An optimized max norm perturbation within an \epsilon bound when added to the sample, the logit distribution is similar to the original sample logit distribution. (2) A random noise was chosen from uniform, Gaussian, or Bernoulli. The authors discuss how the first noise leads the poisons away from the sharp regions and the random noise smooths the loss. The authors show the efficacy of the method against several adaptive attacks. The authors also show that the perturbations generated in this method can transfer across the architectures. Even though the method’s performance is comparable to a few previous defenses, the proposed method is significantly faster.

**Questions:**

Did you try evaluating models on other types of poisoning attacks like Patch-based?

**Strengths And Weaknesses:**

Strengths:
- The technique is simple to understand and use and fast.
- The paper is well-written and easy to follow.
- The investigations seem thorough.

Weakness:
- L129-135, this paragraph says, a high std of KL term means higher sharpness. Can you provide intuition for this? Or cite any previous work discussing this.
- L136 – I think a citation is needed for this statement.

---

> ### Author Response · Authors · 2022-08-02
> **Response to Reviewer tsU4**
>
> We thank the reviewer for the valuable feedback.
>
> 1. **Types of attacks**: We emphasize that our defense is designed to break **visually imperceptible** poisoning attacks, which craft poisons by adding small bounded perturbation to a subset of training data. This category includes targeted triggerless poisoning attacks, namely Witches’ Brew, Bullseye Polytope, Feature Collision, and hidden-trigger backdoor attacks, namely Sleeper Agent and HTBD. This category is inherently the most difficult to detect as the poisons are invisible to human eyes, and the test accuracy is not affected. At the same time, they effectively change the prediction of a particular target example (triggerless targeted), or examples containing a particular patch (hidden-trigger backdoor) at test time. This is in contrast to (potentially dirty-label) triggered backdoored attacks which insert a visible fixed patch into an image, and can be identified by careful visual inspections.
>
> 2. **Std of KL (L129-135)**: If the gradients of randomly sampled points in a small ball around an example has a high variance, it shows the region is not smooth. Indeed in a smooth loss region, the gradients do not change much.
>
> 3. **Sharpness of loss (L136)**: This is our observation. Targeted poisoning attacks change the loss landscape by adding perturbations bounded by $\epsilon$ to a subset of the training data to match the target gradient. Note that the gradient of the *adversarially labeled* target is considerably different from the gradient of the base. Hence, targeted attacks search for a point in $\epsilon$ neighborhood of the base that has a very different gradient. Such a point exists only if there is a sharp loss region around the base example. If such a region exists, the attacker can perturb the base by at most $\epsilon$ to craft an adversarial poison that falls into the sharp region and has a gradient that matches that of the target. The fixed friendly noise part of our method breaks the attacks by maximally perturbing the poisons so that they do not match the target gradient anymore.
>
>     The second part of our method, smoothes out the loss by adding *random varying noise* to all examples. In doing so, it eliminates the sharp loss regions. Hence, the poisons that were falling in those regions before do not fall in sharp regions anymore (as they do not exist), and their gradients cannot match the target anymore. Fig. 2c shows that the noise added by our method indeed smooths out the loss (the variance of gradients around the base becomes much smaller) and hence breaks targeted poisoning attacks.

---

> ### Author Response · Authors · 2022-08-06
> **Looking forward to the reviewer response**
>
> Since we are approaching the end of the author/reviewer discussion period, we appreciate it if the reviewer can read our clarifications and let us know if there is any remaining point we can clarify further. We believe our new results and clarifications address **all the concerns** and we look forward to any further discussion.

---

### Official Review · Reviewer_Sp5u · 2022-07-11

**Rating:** 5
**Confidence:** 4
**Soundness:** 3 good
**Presentation:** 3 good
**Contribution:** 3 good

**Summary:**

This paper proposes to defend against poisoning attacks by adding specific friendly noise to each training example. The friendly noise is simple, effective, optimized before model training, and fixed during model training. Experiments show its effectiveness in defending against various powerful triggerless and backdoor poisoning attacks.

**Questions:**

- Why use the L2 norm in the penalty term of Equation (3)? Can L1 or Linf norm work?
- When the poisons do not mimic the gradient, is the proposed defense still effective?
- Recent works proposed several methods (such as [1][2]) that can also be applied to defend against various attacks. What are the advantages of this method compared with them?

[1] Li, Yige, et al. "Anti-backdoor learning: Training clean models on poisoned data." Advances in Neural Information Processing Systems 34 (2021): 14900-14912.

[2] Wu, Dongxian, and Yisen Wang. "Adversarial neuron pruning purifies backdoored deep models." Advances in Neural Information Processing Systems 34 (2021): 16913-16925.


**Limitations:**

The method only applies to attacks that rely on sharp loss regions such as Witches’ Brew and Gradient Matching. The experiments are only conducted on CIFAR-10.

**Strengths And Weaknesses:**

#### Pros
- This paper is motivated by the observation that powerful poisoning attacks exploit sharp loss regions to craft adversarial perturbations, which is reasonable.
- The proposed method is very simple and easy to implement.
- The visualization of friendly noises is interesting. They seem to have some semantic information.

#### Cons
- The proposed defense relies on the assumption that poisoning attacks craft the poisons to mimic the gradient of the adversarially labeled target. The authors should discuss whether there are attacks with different mechanisms. When applied to such attacks, is the proposed defense still effective?
- The authors criticize that majority of the existing methods are attack specific. However, the proposed method is still specific to the considered attacks that rely on Equation (2).
- The authors criticize that state-of-the-art defense methods are often so expensive that can hardly be applied to even medium-sized datasets. However, experimentally, the authors only verify their method on CIFAR-10. Thus, such criticism can also be used in the proposed method.

---

> ### Author Response · Authors · 2022-08-02
> **Response to Reviewer Sp5u (2/2)**
>
> 4. **Larger experiments**: We compare the run-time of our method with existing defenses. Here we report larger experiments on TinyImagenet:
>
> Table: TinyImageNet/ResNet34 - Gradient Matching (10 datasets) using FRIENDs-B defense, 200 epochs. Note that running the SOTA defense AP-0.5 on TinyImageNet/ResNet34 takes 7 hours for 40 epochs, we estimate AP-0.5 for 200 epochs to take at least 30 hours for a single dataset.
>
> Defense|Poison Acc|Test Acc|Runtime (HH:MM)
> ---|---|---|---
> Undefended|60.00%|60.83%|04:01
> FRIENDs-B|0.00%|55.77%|04:28
>
>
> 5. **Other baselines**: We emphasize that existing defenses against triggered attacks, such as “Anti-backdoor learning (ABL) and Adversarial neuron pruning” are specific to triggered backdoor attacks and are particularly designed to **unlearn** the triggers after they are learned by the model. Indeed, they are not effective in defending against targeted triggerless or hidden-trigger backdoor attacks. Our new experiments with ABL [1] below confirms ineffectiveness of ABL against hidden-trigger backdoors (Sleeper Agent). Similarly, Adversarial neuron pruning is only effective against triggered backdoors as the specific pattern in the patch is learned by particular neurons. This is not the case for targeted triggerless and hidden-trigger backdoor attacks that generate poisons under bounded perturbations. On the other hand, our defense prevents targeted triggerless attacks and hidden-trigger backdoors from being learned in the first place. Indeed, our defense (due to being effective and light-weight) can be applied simultaneously with methods such as ABL to defend against both triggerless and triggered poisoning attacks.
>
> Table: ABL, implementation adapted from https://github.com/bboylyg/ABL.
> Original ABL did not report results on ResNet-18 nor Sleeper Agent attack. We directly substituted WRN-16-1 with ResNet-18, and the backdoor attack with Sleeper Agent in their code.
>
> Sleeper Agent attack (ξ = 16)|Poison Acc|Test Acc
> |---|---|---|
> Undefended|91.72%|93.36%
> ABL (After Unlearning)|63.90%|75.14%
> FRIENDs-B|31.20%|91.31%
>
> [1] Li, Yige, Xixiang Lyu, Nodens Koren, Lingjuan Lyu, Bo Li, and Xingjun Ma. "Anti-backdoor learning: Training clean models on poisoned data." Advances in Neural Information Processing Systems 34 (2021): 14900-14912.

---

> ### Author Response · Authors · 2022-08-02
> **Response to Reviewer Sp5u (1/2)**
>
> We thank the reviewer for acknowledging the simplicity, effectiveness, and practicality of our method in defending against powerful targeted poisoning attacks.
>
> 1. **Targeted attacks & gradient matching**: We emphasize that our defense is designed to break **visually imperceptible** poisoning attacks, which craft poisons by adding small bounded perturbation to a subset of training data. This category includes targeted triggerless poisoning attacks, namely Witches’ Brew, Meta Poison, Bullseye Polytope, Feature Collision, and hidden-trigger backdoor attacks, namely Sleeper Agent and HTBD. As discussed in Sec. 3, the above category of attacks all work by adding bounded perturbations to the base images to *match the gradient/representation of the adversarially labeled targets or patches sources*. In doing so, the attack minimizes the adversarial loss while minimizing the training loss (Eq. 2). Note that while some of the above attacks, e.g. Meta Poison, do not directly work by matching gradients and instead aim to directly solve Eq. 1, as a bilevel optimization problem. However, the generated poisons based on Eq. 1 match the target gradient, as any solution for Eq. 1 is also a solution for Eq. 2. We note that **our defense is not restricted to gradient matching**, and indeed our method is effective against all *attacks that generate poisons under bounded perturbations*. Our discussion of the gradient matching is mainly to provide intuition on the effect of our defense method on training and optimization. We will highlight and emphasize this in our revised version.
>
> 2. **Attack specific defense**: We emphasize that our defense is indeed not attack specific and is effective against all different types of *visually imperceptible poisoning attacks*. Note that Eq. 2 is the objective of various types of targeted poisoning attacks, including Witches Brew, BullEyes Polytope, Sleeper Agent, Feature Collision, etc. The friendly noise is indeed **data specific**, but is agnostic against the type of targeted attacks. As we showed in Sec. 4.3, our friendly noise is also transferable, i.e. it can be generated on a model and be applied to defend other models.
>
> 3. **L1, L2 and L_inf**: L1 and L_inf can be also used in Eq. 3. L_inf only cares about the largest element in the noise vector, so it adds larger perturbations, while L_1 cares about absolute values hence it adds smaller perturbations. L2 penalizes large noise elements more than L1, so it adds slightly smaller perturbations. Our new experiments with L_1 and L_inf show that different norms result in similar poison and test accuracy. We thank the reviewer and will add the discussion to our revised version.
>
> Table: CIFAR-10 - Gradient Matching (ξ = 16) Performance is similar across L1/L2 norms. Here we use FRIENDs-B with the same defense settings as Table 1.
>
> Norm|Poison Acc|Test Acc
> ---|---|---
> L2|0.00%|91.52%
> L1|0.00%|91.50%
> L_inf|0.00%|91.37%

---

> ### Author Response · Authors · 2022-08-06
> **Looking forward to the reviewer response**
>
> Since we are approaching the end of the author/reviewer discussion period, we appreciate it if the reviewer can read our clarifications and let us know if there is any remaining point we can clarify further. We believe our new results and clarifications address **all the concerns** and we look forward to any further discussion.

---

### Official Review · Reviewer_PZZ3 · 2022-07-11

**Rating:** 3
**Confidence:** 4
**Soundness:** 2 fair
**Presentation:** 2 fair
**Contribution:** 2 fair

**Summary:**

This paper claim that poisoned data should lie around the sharp loss regions. Therefore, this paper proposes to add "friendly" noise to drift poisoned data away from the sharp loss regions.

**Questions:**

1 There exist (robust) unlearnable examples [1][2], which can degrade the learning performance.
Could the friendly noise make the (robust) unlearnable example learnable?
[1]UNLEARNABLE EXAMPLES: MAKING PERSONAL DATA UNEXPLOITABLE, in ICLR 2021
[2] ROBUST UNLEARNABLE EXAMPLES: PROTECTING DATA AGAINST ADVERSARIAL LEARNING, in ICLR 2022

2 Could the poison attacker also design adaptive poisoning attacks that make the fixed-designed friendly noise ineffective?

3 Suppose the training set is clean and benign, would the friendly noise degrades the training performance?

**Limitations:**

No. This paper did not discuss the limitations and potential negative societal impact of their work.

**Strengths And Weaknesses:**

++ I think it is practical to post-process the training set (e.g., removing the adversarial or noisy data) before feeding it to the learning algorithm.
Falling in this catogory, this paper proposes adding "friendly noise + random noise" to each training data before updating model parameters.

++ The friendly noises reinforce and highlight salient (robust) features of each training data, suppressing those unnoticeable (non-robust) features that could subvert the predictions. Due to this property of friendly noise, I suspect it will make the triggered-based backdoor attacks even worse because the triggers themselves are salient features.

--A prior study exists that "Anti-Backdoor Learning: Training Clean Models on Poisoned Data" in NeurIPS 2021.
The prior study seems to defend against poisoning attacks without degrading the training performance.
Are there empirical comparisons between the friend noise with those effective and recent prior studies?

--This is a very strong claim that "friendly noise" makes it extremely difficult for an adaptive attacker to break the defense.
The empirical evaluations in this paper are only based on some weak poisoning attacks, such as clean-label targeted/transfer attacks. I encourage the authors to evaluate the fixed-designed friendly noise on the stronger untargeted attacks, such as unlearnable examples and backdoor triggers.

--The claim "powerful poisons fall in the sharp loss regions" is unsubstantiated. First, the loss should be the function of the model, data, and label. What is the exact meaning of that "powerful poisons fall in the sharp loss regions"? Second, the insights and arguments in Section 3.1 are under-supported. For example, why cannot the learning learn and adapt the poisoned data and make the loss smaller and flatter around the poisoned data?

---

> ### Author Response · Authors · 2022-08-02
> **Response to Reviewer PZZ3 (3/3)**
>
> 5. **Accuracy on clean data**: our defense shouldn’t degrade the performance on clean data more than that of poisoned data. We confirm experimentally in the 40 epoch setting. This is in contrast to defenses like DP-SGD and Activation Clustering that considerably degrade the performance.
>
>
> |Defense|Test Acc on clean data | Test acc on poisoned data (Gradient Matching)|
> |---|:------------|:----------|
> None|93.63%|93.50%
> FRIENDs-B|90.87%|91.52%
> FRIENDs-G|91.46%|91.50%
> FRIENDs-U|92.22%|91.91%
> DP-SGD|76.96%|75.70%
> Activation Clustering|87.97%|87.69%
>
> 6. **Sharp loss regions**: Sharp loss region is referred to a neighborhood where small changes in the parameters w result in large change to the loss $L$ and hence its gradient, i.e. $0 \leq |L(w_t+\epsilon)-L(w_t)| \leq C_1$ for a small $\epsilon$ and a large constant $C_1$. This is in contrast to (Lipschitz) smooth regions where the loss does not change considerably, i.e. $0 \leq |L(w_t+\epsilon)-L(w_t)| \leq C_2$ for a small $C_2$. As correctly mentioned by the reviewer, the loss is a function of the model, **data**, and labels. Targeted poisoning attacks change the loss landscape by adding perturbations bounded by \epsilon to a subset of the training **data** to match the target gradient. Note that the gradient of the *adversarially labeled* target is considerably different from the gradient of the base. Hence, targeted attacks search for a point in $\epsilon$ neighborhood of the base that has a very different gradient. Such a point exists only if there is a sharp loss region around the base example. If such a region exists, the attacker can perturb the base by at most \epsilon to craft an adversarial poison that falls into the sharp region and has a gradient that matches that of the target. The fixed friendly noise part of our method breaks the attacks by maximally perturbing the poisoned **data** so that they do not match the target gradient anymore. The second part of our method, smoothes out the loss by adding *random varying noise* to all training **data**. In doing so, it eliminates the sharp loss regions. Hence, the poisons that were falling in those regions before do not fall in sharp regions anymore (as they do not exist), and their gradients cannot match the target anymore. Fig. 2c shows that the noise added by our method indeed smooths out the loss (the variance of gradients around the base becomes much smaller), and hence breaks targeted poisoning attacks.
>
> 7. **Social impact**: In Sec. 5 (conclusion), we discussed the scope of our work and the types of attacks we can address. We are not aware of any negative social impact of our work as we believe defending against targeted attacks is very important for the society, but we are happy to include any suggestion the reviewer may have.
>
>
> [1] Li, Yige, Xixiang Lyu, Nodens Koren, Lingjuan Lyu, Bo Li, and Xingjun Ma. "Anti-backdoor learning: Training clean models on poisoned data." Advances in Neural Information Processing Systems 34 (2021): 14900-14912.
>
> [2] Huang, Hanxun, Xingjun Ma, Sarah Monazam Erfani, James Bailey, and Yisen Wang. "Unlearnable examples: Making personal data unexploitable." arXiv preprint arXiv:2101.04898 (2021).
>
> [3] Fu, Shaopeng, Fengxiang He, Yang Liu, Li Shen, and Dacheng Tao. "Robust unlearnable examples: Protecting data privacy against adversarial learning." In International Conference on Learning Representations. 2021.

---

> ### Author Response · Authors · 2022-08-02
> **Response to Reviewer PZZ3 (2/3)**
>
> 3. **Unlearnable examples**: We note that unlearnable examples [2,3] are inherently different from targeted poisoning attacks. Unlearnable examples perturb **all training examples** to *maximize overfitting*. In doing so, **the data owners protect privacy** and prevent unauthorized learning from their data. On the other hand, targeted and backdoor data poisoning attacks only poison **a subset** of training examples to modify models’ behavior on *particular test time examples*. Indeed, it may not be a reasonable assumption for an adversary to poison all the training examples. Nevertheless, we ran new experiments showing the effect of our defense method on unlearnable examples [2]. We note that here only the second part of our method, i.e. random varying noise (with a large magnitude of $\epsilon=48, 56$ *which is not used in our paper*) is relevant to prevent overfitting and increase accuracy of learning from unlearnable examples. We emphasize that this is indeed not the focus of our method. Instead, we aim at protecting models from triggerless targeted and hidden trigger backdoor poisoning attacks while preserving the test accuracy.
>
> Table: Unlearnable Examples - FRIENDs-G with larger Gaussian noise component (total $\epsilon=64$)
> |   |Test Acc|
> |---|---|
> Undefended|15.96%
> FRIENDs-G ($\epsilon_{noise}=48$)|45.75%
> FRIENDs-G ($\epsilon_{noise}=56$)|51.54%
>
>
> 4. **Adaptive attacks**: We believe that an adaptive attack against our defense is extremely difficult due to the following reasons. First, note that it is nontrivial, or even almost computationally impossible, for an attacker to be fully adaptive to our friendly noise. Adaptively generated poisons would have to be optimized based on the friendly noise generated. But, the friendly noise generated will in fact be a function of the adaptively generated poisons. In other words, any adaptive poisoned dataset $D_{ap}$ has to be a function of itself, i.e. we require $D_{ap} = g(f(D_{ap}))$ where $f(.)$ gives the friendly noise and $g(.)$ generates the attack. Considering that $f(.)$ consists of many steps of SGD, and $g(.)$ itself is usually equally complex, directly optimizing for such a dataset is computationally prohibitive.
>
>     Second, note that the second part of our method, i.e. random varying noise, prevents any adaptive attack from being successful even with the knowledge of the fixed noise (as discussed in Sec 3.4). Adaptive attacks incorporate an existing defense in the poison generation process (Eq. 1, 2). However, as our method adds random varying noise to the examples, it becomes prohibitively expensive for an adaptive attacker to incorporate all different combinations of random noise (in addition to the fixed noise) while crafting poisons to adaptively attack our defense.
>
>     We ran new experiments showing that incorporating random noise in addition to the fixed noise (assuming the attacker is aware of it) during poison generation cannot fully break our defense method. We modified the differentiable data augmentation component of Sleeper Agent attack algorithm to include randomly sampled Bernoulli noise of $\epsilon=16$ (the same as what we use in our defense) in row 2. In row 3, we further added the fixed friendly noise generated from selected prior runs to the attacker’s augmentation procedure. Due to time constraints, we only ran 10 datasets for rows 2-3, we will update with the full 24 datasets. We observe that despite attacks being adapted to Bernoulli and friendly noise, poison accuracies are all within standard deviation of one another, and the adaptive attacks cannot succeed.
>
> Table: Sleeper Agent attack (“adaptive”) generated with random Bernoulli noise, with ξ = 16 (averaged over 10 datasets). Even though attack is generated “adaptively” with Bernoulli $\epsilon=16$ noise, it still fails to be robust against Bernoulli noise defense.
> ||Undefended|         |Defended (FRIENDs-B)|     |
> |---|---|---|---|---|
> |Attack|Poison Acc|Test Acc|Poison Acc|Test Acc
> Sleeper Agent|$83.48 \pm 7.58$|$91.56 \pm 0.19$|$21.52 \pm 8.39$|$89.76 \pm 0.30$
> Sleeper Agent + Bernoulli Noise|$81.30 \pm 12.33$|$91.40 \pm 0.21$|$29.57 \pm 14.43$|$89.77 \pm 0.41$
> Sleeper Agent + Friendly and Bernoulli noise|$77.18 \pm 13.27$|$91.71 \pm 0.33$|$31.56 \pm 14.70$|$88.48 \pm 0.41$

---

> ### Author Response · Authors · 2022-08-02
> **Response to Reviewer PZZ3 (1/3)**
>
> We thank the reviewer for the valuable feedback and acknowledging the practicality of our method in breaking poisoning attacks. Below, we clarify on the type of attacks our work targets, comparison to existing defense, adaptive attacks, and sharp loss argument:
>
> 1. **Type of attacks**: There seems to be a misunderstanding of the types of attacks we target in our work. As *defined* in the first sentence of the abstract, and explicitly mentioned in the related work, first sentence of Sec. 3 and Eq. 1, and conclusion, and is acknowledged by reviewer Nmev, our defense is designed to break (clean-label) **visually imperceptible** poisoning attacks, which craft poisons by adding small **bounded perturbations** to a subset of training data to change prediction of particular target examples at test time. This category includes targeted triggerless poisoning attacks, namely Withche’s Brew, Bullseye Polytope, Feature Collision, and hidden-trigger backdoor attacks, namely Sleeper Agent and HTBD. This category is inherently the most difficult to detect as the poisons are invisible to human eyes, and the test accuracy is not affected. This makes them one of the most concerning threats to deep learning systems. Note that this category is separate from triggered backdoored attacks that insert a mostly visible fixed patch into an image. The inserted patch is indeed not created under bounded perturbations, and can often be identified by careful visual inspections.
>
>     We disagree that triggered backdoors are more powerful than triggerless attacks. Triggered backdoors need separate treatments that are indeed not effective against triggerless and hidden-trigger data poisoning attacks (as confirmed by our new experiments below).
>
>     We acknowledge and apologize that this is not clearly specified in some sentences in the abstract and intro, which may have caused a confusion. We thank the reviewer for pointing out this confusion and have modified the title of the paper and sections, abstract, and intro to clarify. We hope the reviewer can revisit the score based on our clarifications.
>
> 2. **Comparison to existing defenses**: We have the comparison with all existing baselines for defending against **visually imperceptible** poisoning attacks in the original paper, which confirms the superiority of our method. We emphasize that existing defenses against triggered attacks, such as “Anti-backdoor learning (ABL)” [1] are not effective in defending against targeted triggerless or hidden-trigger backdoor attacks as we confirm by our new experiments below. Note that ABL is specific to triggered backdoor attacks, and is particularly designed to *unlearn* the triggers after they are learned by the model, and needs to know the number of poisoned examples. On the other hand, our defense prevents targeted triggerless attacks and hidden-trigger backdoors from being learned in the first place, and does not rely on the information about the number of poisoned examples. Indeed, our defense (due to being effective and light-weight) can be applied simultaneously with methods such as ABL to defend against both triggerless and triggered poisoning attacks.
>
>     The table below shows ABL being applied on Sleeper Agent attack (hidden-trigger backdoor), which results in a very high poison success rate while significantly harming test accuracy. This shows that when applied to such attacks, ABL is ineffective in identifying poisons, even with the knowledge of the number of poisons. The fundamental assumption made by ABL that poisons can be identified based on losses in the first few epochs of training does not hold here. Thus, ABL drops many clean examples.
>
> Table: ABL, implementation adapted from https://github.com/bboylyg/ABL.
> Original ABL did not report results on ResNet-18 nor Sleeper Agent attack. We directly substituted WRN-16-1 with ResNet-18, and the backdoor attack with Sleeper Agent in their code.
>
> Sleeper Agent attack (ξ = 16)|Poison Acc|Test Acc
> |---|---|---|
> Undefended|91.72%|93.36%
> ABL (After Unlearning)|63.90%|75.14%
> FRIENDs-B|31.20%|91.31%

---

> ### Author Response · Authors · 2022-08-06
> **Looking forward to the reviewer response**
>
> Since we are approaching the end of the author/reviewer discussion period, we appreciate it if the reviewer can read our clarifications and let us know if there is any remaining point we can clarify further. We believe our new results and clarifications address **all the concerns** and we look forward to any further discussion.

---

> ### Comment · Reviewer_PZZ3 · 2022-08-08
> **Thanks for authors' responses.**
>
>
> Dear Authors,
>
> Many thanks for responding to my concerns. However, my concerns on "whether the friendly noise works on untargeted poison attacks/backdoor triggers," "paper's presentation," and "paper's practicality" remain.
>
> 1 In rebuttal, the authors “emphasized” that the friendly noise ONLY tackles the targeted imperceptible poisoning attacks.
> I am fine with this argument; however, I highlight that this could significantly limit the practicality of this paper.
>
> 2 “Anti-backdoor learning (ABL)” [1] can also deal with invisible triggers, i.e., unnoticeable noise added to the input data. Since the poisoning noise is unnoticeable, it is hard to distinguish the harm on a targeted data point or many untargeted data points. I guess the authors may misunderstand my initial comments.
> ABL tackles the backdoor triggers, which is a different setting from this paper. I would like to know whether friendly noise works under ABL's setting.
>
> 3 Again, my arguments above apply to the unlearnable examples [2,3]. It seems that eps = 48 and 56 is too large. It may signify that friendly noise cannot work well under the setting of untargeted attacks and backdoor triggers.
>
> 4 " Suppose the training set is clean and benign, would the friendly noise degrades the training performance?" Sorry for my confusion if my question is unclear. Let me rephrase my question: Suppose we have a clean dataset that is poisoning free, would the friendly noise disturb the signals and cause harm to the test accuracy? It is because we will never know whether the dataset is poisoned or not beforehand. I would like to ensure there is no/little side effect of the friendly noise.
>
> 5 "The claim 'powerful poisons fall in the sharp loss regions' is unsubstantiated."
> Thanks for the explanations for this. However, I still feel the explanations are superficial. Maybe, the authors could rigorously (e.g., mathematically) connect the weight perturbations and data perturbations w.r.t. loss value (and even the true risk).
>
> Based on my above concerns, I am so sorry that I cannot raise my scores and uphold your paper as other reviewers did.
>
> Best, \
> Reviewer PZZ3

---

> > ### Author Response · Authors · 2022-08-09
> > **Further clarifications**
> >
> > We thank the reviewer for the new feedback. We believe all the confusion are already addressed by our initial answer to your comments (**in 3 parts, including several new experiments**), but you may have missed them. We clarify below:
> >
> > 1. Please note that as we mentioned in our answer, our method tackles (1) imperceptible targeted poisoning attacks and (2) untargeted hidden-trigger backdoor attacks (not only imperceptible targeted attacks). This indeed does not limit the practicality of our approach. Due to the importance of the problem (explained in our initial answer), many existing defenses aim to solely defend against such attacks - [1,2,3,4,5,6,7,8,9,10,11]. Our method is indeed superior to all such existing methods.
> >
> > 2. We respectfully disagree with the reviewer. Indeed, **ABL is not designed to deal with invisible attacks and does not claim to do so** (patches are indeed not invisible)! Please see the table in our initial answer where **we ran new experiments showing that ABL fails** against the backdoor attack “Sleeper Agent” [12] (the only effective hidden-trigger backdoor attack method for from scratch training). We show that ABL drops the accuracy by 20% and cannot prevent poisoning. As we explained in our initial answer, **our method can indeed be applied with ABL simultaneously** to defend against visible backdoors and invisible targeted and backdoor attacks. This is unlike most of the existing methods that cannot be applied with other defenses simultaneously [2,3,4,5,7,8,9,10].
> >
> >     We ask the reviewer for additional clarification on the following statement: “Since the poisoning noise is unnoticeable, it is hard to distinguish the harm on a targeted data point or many untargeted data points.” We did not understand what this sentence meant.
> >
> > 3. As we explained in our original answer (see Part 2/3), **unlearnable examples are not data poisoning attacks**. They aim to protect privacy. Unlearnable examples add noise to **all the training data**, and does not let the model learn generalizable information from the very beginning. This is indeed against the definition of data invisible poisoning attacks in which adversary **poisons a subset of training data without changing the test accuracy**. Hence, we do not think the applicability of our method is relevant to this setting. Indeed, none of the existing defenses against data poisoning attacks are applicable to unlearnable examples.
> >
> > 4. We ran new experiments in our initial response (see Part 3/3), to confirm that our method does not drop accuracy on clean data (please see table included in original response). We believe the reviewer might have missed our comments and our additional experiments.
> >
> > 5. We respectfully disagree that our claim on the sharpness of the loss is superficial, and we believe that we have substantiated our claim with sufficient rigor. As we explained in our original answer (Part 3/3), for poisoning to succeed, the gradient obtained from training on poisons must match the adversarial gradient (from the target). Adversarial gradient directions are naturally very different from clean directions. Hence, in order for such adversarial gradient directions to be generated, the loss region corresponding to these adversarial directions has to be sufficiently sharp to significantly alter training dynamics towards this highly different direction. Under Lipschitz assumption with respect to weights, bounded perturbations to weights result in bounded perturbations to the loss. I.e. L is (locally) Lipschitz in neighborhood $\mathcal{U}$ if there exists (real) constant $K$ s.t. $|L(w_1) - L(w_2)| \leq K|w_1 - w_2| \ \forall \ w_1, w_2 \in \mathcal{U}$. In particular, since loss is locally bounded, gradients are locally bounded as well. Here, smoothness and sharpness of loss regions are mathematically defined by the constants that bound this change in loss as a result of weight perturbations. By smoothing (i.e. smaller gradient/Lipschitz constant) the local region, we remove sharp loss regions (i.e. larger gradients/Lipschitz constant) required for data poisoning to succeed. We also experimentally verified this in Fig 2c.
> >
> > We hope the reviewer can reconsider their evaluation after our clarifications. Thank you.

---

> > > ### Author Response · Authors · 2022-08-09
> > > **References**
> > >
> > > [1] Yang, Yu, Tian Yu Liu, and Baharan Mirzasoleiman. "Not All Poisons are Created Equal: Robust Training against Data Poisoning." In International Conference on Machine Learning, pp. 25154-25165. PMLR, 2022.
> > >
> > > [2] Peri, N., Gupta, N., Huang, W. R., Fowl, L., Zhu, C., Feizi, S., Goldstein, T., and Dickerson, J. P. Deep k-nn defense against clean-label data poisoning attacks. In European Conference on Computer Vision, pp. 55–70. Springer, 2020.
> > >
> > > [3] Tao, L., Feng, L., Yi, J., Huang, S.-J., and Chen, S. Better safe than sorry: Preventing delusive adversaries with adversarial training. Advances in Neural Information Processing Systems, 34, 2021
> > >
> > > [4] Madry, A., Makelov, A., Schmidt, L., Tsipras, D., and Vladu, A. Towards deep learning models resistant to adversarial attacks. In International Conference on Learning Representations, 2018.
> > >
> > > [5] Bryant Chen, Wilka Carvalho, Nathalie Baracaldo, Heiko Ludwig, Benjamin Edwards, Taesung Lee, Ian Molloy, and Biplav Srivastava. Detecting backdoor attacks on deep neural networks by activation clustering. In SafeAI@ AAAI, 2019
> > >
> > > [6] Yansong Gao, Change Xu, Derui Wang, Shiping Chen, Damith C. Ranasinghe, and Surya Nepal.  Strip: A defence against trojan attacks on deep neural networks. In Proceedings of the 35th Annual Computer Security Applications Conference, ACSAC ’19, page 113–125, New York, NY, USA, 2019. Association for Computing Machinery.
> > >
> > > [7] Bolun Wang, Yuanshun Yao, Shawn Shan, Huiying Li, Bimal Viswanath, Haitao Zheng, and Ben Y Zhao. Neural cleanse: Identifying and mitigating backdoor attacks in neural networks. In 2019 IEEE Symposium on Security and Privacy (SP), pages 707–723. IEEE, 2019.
> > >
> > > [8] Geiping, J., Fowl, L., Somepalli, G., Goldblum, M., Moeller, M., and Goldstein, T. What doesn’t kill you makes you robust (er): Adversarial training against poisons and backdoors. 2021a
> > >
> > > [9] Hong, S., Chandrasekaran, V., Kaya, Y., Dumitras¸, T., and Papernot, N. On the effectiveness of mitigating data poisoning attacks with gradient shaping. arXiv preprint arXiv:2002.11497, 2020.
> > >
> > > [10] Tran, B., Li, J., and Madry, A. Spectral signatures in backdoor attacks. In Advances in Neural Information Processing Systems, pp. 8000–8010, 2018.
> > >
> > > [11] Eitan Borgnia, Valeriia Cherepanova, Liam Fowl, Amin Ghiasi, Jonas Geiping, Micah Goldblum, Tom Goldstein, and Arjun Gupta. Strong data augmentation sanitizes poisoning and backdoor attacks without an accuracy tradeoff. In ICASSP 2021-2021 IEEE International Conference on Acoustics, Speech and Signal Processing (ICASSP), pages 3855–3859. IEEE, 2021
> > >
> > > [12] Souri, H., Goldblum, M., Fowl, L., Chellappa, R., and Goldstein, T. Sleeper agent: Scalable hidden trigger backdoors for neural networks trained from scratch. arXiv preprint arXiv:2106.08970, 2021.

---

### Meta-Review · Area_Chair_xw8t · 2022-08-28

**Recommendation:** Accept
**Confidence:** Certain

**Metareview:**

This paper proposes a poisoning defense that unlike existing methods breaks various types of poisoning attacks with a small drop in the generalization. The key claim is that attacks exploit sharp loss regions to craft adversarial perturbations which can substantially alter examples' gradient or representations under small perturbations. The authors then propose to generate noise patterns which maximally perturb examples while minimizing performance degradation.

I think the framing of this paper is very important and the authors have done a good job at it. They are claiming to have a defense that is non attack-specific as long as it is restricted to the class of attacks involving visually imperceptible inputs. I believe this claim, if substantiated, to be of sufficient significance to the NeurIPS community.

Unfortunately, I noticed that the reviewers largely did not respond to the author rebuttal, other than PZZ3. PZZ3's main concerns were with lack of novelty with respect to the Anti-Backdoor Learning paper, different settings (untargeted, backdoor triggers), and substantiation for the sharp loss hypothesis. Having read carefully the authors' rebuttal to these, I believe the authors have a done a good job of alleviating the concerns and/or misunderstandings. For example I've read the ABL paper and agree it is dealing with a different setting. It was nice to see that the authors actually did experiments to show that ABL is not effective in this setting and vice versa.

Reviewer Sp5u was concerned with the attack-specificity of the defense to which the authors rebutted appropriately that is is data specific but not attack-specific as long as it is restricted to the class of attacks involving visually imperceptible inputs.

As far as I can tell, there were no other strong concerns.

Based on my own assessment, I believe that the central claim of the paper has sufficient evaluations to support it. The attacks considered are highly varied in their techniques and are also recent and SOTA.

I therefore recommend accept.

**Award:**

No

---

### Decision · Program_Chairs · 2022-09-14

Accept